



# Shortwave Array Spectroradiometer-Hemispheric (SAS-He): Design and Evaluation

Evgueni Kassianov[1], Connor J. Flynn[2], James C. Barnard[3], Brian D. Ermold[1], Jennifer M. Comstock[1]

[1]Pacific Northwest National Laboratory, Richland, WA, USA
[2]School of Meteorology, University of Oklahoma, Norman, OK, USA
[3]University of Nevada, Reno, NV, USA

*Correspondence to*: Evgueni Kassianov (Evgueni.Kassianov@pnnl.gov)

**Abstract.** A novel ground-based radiometer, referred to as the Shortwave Array Spectroradiometer-Hemispheric (SAS-He),
is introduced. This radiometer uses the shadow band technique to report total irradiance and its direct and diffuse components frequently (every 30 sec) with continuous spectral coverage (350-1700 nm) and moderate spectral (~2.5 nm ultraviolet/visible, and ~6 nm shortwave-infrared) resolution. The SAS-He's performance is evaluated using integrated datasets collected over coastal regions during three field campaigns supported by the U.S. Department of Energy's (DOE's) Atmospheric Radiation Measurement (ARM) Program, namely (1) Two-Column Aerosol Project (TCAP; Cape Cod, Massachusetts), (2)
Tracking Aerosol Convection Interactions Experiment (TRACER; in and around Houston, Texas), and (3) Eastern Pacific Cloud Aerosol Precipitation Experiment (EPCAPE; La Jolla, California). We compare (i) aerosol optical depth ($AOD$) and total optical depth ($TOD$) derived from the direct irradiance, (ii) the diffuse irradiance and direct-to-diffuse ratio ($DDR$) calculated from two components of the total irradiance. As part of the evaluation, both $AOD$ and $TOD$ derived from the SAS-He direct irradiance are compared to those provided by collocated Cimel sunphotometer (CSPHOT) at five (380, 440, 500,
675, 870 nm) and two (1020, 1640 nm) wavelengths, respectively. Additionally, the SAS-He diffuse irradiance and $DDR$ are contrasted with their counterparts offered by a collocated Multi-Filter Rotating Shadowband Radiometer (MFRSR) at six (415, 500, 615, 675, 870, 1625 nm) wavelengths. Overall, reasonable agreement is demonstrated between the compared products despite the challenging observational conditions associated with varying aerosol loadings and diverse types of aerosols and clouds. The $AOD$- and $TOD$-related values of root-mean-square error are within the expected measurement uncertainty of $AOD$
25   (0.01-0.02).

## 1 Introduction

Clouds and atmospheric aerosols are important drivers of the Earth radiation budget (Voigt et al., 2021; Li et al., 2022). They have unique fingerprints across a wide spectral range, which includes ultraviolet (UV), visible (VIS), near-infrared (NIR) and shortwave-infrared (SWIR) spectral bands. Moreover, cloud droplets and aerosol particles with a wide range of sizes and
shapes modify distinctly the angular distribution of the scattered solar radiation (Hansen and Travis, 1974; Kokhanovsky,



2004; Yang et al., 2013). Thus, advanced cloud and aerosol retrievals over different surface types involve both multi-spectral and multi-angular measurements from surface, air, and space (Chen et al., 2022; Puthukkudy et al., 2020; Matar et al., 2023; Michalsky and Kiedron, 2022). To illustrate, the aircraft-based sensor named Spectrometer for Sky-Scanning, Sun-Tracking Atmospheric Research (4STAR; wavelength range: 350–1650 nm) with sky-scanning and hyperspectral abilities has offered

valuable information on above-cloud aerosol optical depth ($AOD$) (LeBlanc et al., 2020), while solar irradiances measured by aircraft-based Solar Spectral Flux Radiometer (SSFR; wavelength range: 350–2100 nm)  have been used to provide spectral surface albedo (Coddington et al., 2008) and cloud spectral absorption (Kindel et al., 2011), which is a function of cloud optical thickness and droplet effective radius. The combined 4STAR and SSFR airborne measurements have been utilized to derive heating rate profiles over a climate-important region (Cochrane et al., 2022).

Airborne measurements can infer valuable properties of clouds and aerosols with high temporal resolution. However, these episodic measurements represent a relatively short period (e.g., several weeks) and a given location (LeBlanc et al., 2020; Cochrane et al., 2022). Conversely, satellite observations have been used successfully to extract a wealth of information about clouds and aerosols near-globally (Platnick et al., 2017; Gumber et al., 2023). Nevertheless, these observations occur infrequently (typically several times a day) and do not capture the diurnal cycle. The airborne and satellite measurements can

be supplemented substantially by the long-term ground-based radiation data collected with high temporal resolution at multiple sites with worldwide locations (Remer et al., 2023). For example, Aerosol Robotic Network (AERONET) Program with world-wide distributed sites has provided columnar $AOD$s at seven wavelengths (380, 440, 500, 675, 870, 1020, 1640 nm) from the direct-beam irradiance measured by Cimel sunphotometers (CSPHOT; Holben et al., 1998; Giles et al., 2019). Similar to the AERONET sunphotometers, Multi-Filter Rotating Shadowband Radiometers (MFRSRs) supported by the U.S. Department of

Energy's (DOE's) Atmospheric Radiation Measurement (ARM) Program (Sisterson et al., 2016; Miller et al., 2016) and the National Oceanic and Atmospheric Administration's (NOAA's) Surface Radiation budget network (SURFRAD; Augustine et al., 2008) have provided $AODs$ at five wavelengths (415, 500, 615, 675, 870 nm) for many locations from the direct irradiances. These irradiances have been obtained from the MFRSR-measured total and diffuse solar irradiances. Additionally, the MFRSR data have been used to derive aerosol, cloud, and surface properties (Riihimaki et al., 2021), and to quantify the spectrally

resolved radiative impact of clouds (Kassianov et al., 2011). Recently, ARM has added a new channel at 1625 nm wavelength to the ARM-supported MFRSRs.

The limited number of wavelengths coupled with a quite narrow spectral range of the MFRSR prevent improved retrievals of cloud, aerosol, and surface characteristics, and thus preclude advanced understanding of complex cloud-aerosol-surface interactions (Barthlott et al., 2022; Calderón et al., 2022). To address the outlined limitation, ARM developed a hyperspectral

shortwave radiometer, called the Shortwave Array Spectroradiometer-Hemispheric (SAS-He), that has collected data since 2011. This ground-based radiometer is a next-generation of the MFRSR with increased spectral coverage (350-1700 nm) and hyperspectral ability. Here we illustrate its performance by taking advantage of an integrated dataset collected by collocated ground-based sensors over coastal regions during three campaigns: (1) Two-Column Aerosol Project (TCAP; Cape Cod, Massachusetts) (Berg et al., 2016), (2) Tracking Aerosol Convection Interactions Experiment (TRACER; in and around



Houston, Texas) (Jensen et al., 2022) and (3) Eastern Pacific Cloud Aerosol Precipitation Experiment (EPCAPE; La Jolla, California) (Russell et al., 2021). The following four sections cover the SAS-He design and calibration procedures (Sect. 2), a concise description of ground-based data collected during three campaigns (TCAP, TRACER, and EPCAPE) (Sect. 3), evaluation of the SAS-He $AOD$, total optical depth ($TOD$), direct-to-diffuse ratio ($DDR$), diffuse irradiance (Sect. 4), and a summary of main results (Sect. 5). It should be emphasized that the spectrally resolved $AODs$ offer a valuable avenue for

estimating aerosol columnar size distributions (e.g., King et al., 1978; Sayer et al., 2012; Kassianov et al., 2021; Torres and Fuertes, 2021). Aerosol intensive properties, including single-scattering albedo and asymmetry parameter, are possible through retrievals combining $AODs$ and $DDRs$ (e.g., Kassianov et al., 2007; Ge et al., 2010). Additionally, the favorable comparisons demonstrated in our paper can be considered as foundational for future activities including improved understanding of changes in photosynthetically active radiation, and refinement of broadband radiation measurements and radiative transfer calculations.

Finally, the potential exists to retrieve cloud properties, such as cloud optical depth and droplet effective radius, by examining wavelength-dependent diffuse irradiance under overcast conditions (e.g., LeBlanc et al., 2015).

## 2 SAS-He design, calibration and corrections

    The SAS-He is the successor to the MFRSR (Hodges and Michalsky, 2016) and the Rotating Shadow band Spectroradiometer

(RSS; Michalsky and Kiedron, 2022). The SAS-He employs the shadow band technique (Wesely, 1982; Harrison et al., 1994) and its design, operation and calibration are borrowed heavily from the predecessors. The corresponding detailed reviews are presented elsewhere (Hodges and Michalsky, 2016; Flynn, 2016; Michalsky and Kiedron, 2022), and here we provide only a brief description sufficient to acquaint the reader with enhanced capabilities of the SAS-He.

### 2.1 Design

Figure 1 shows the major elements of the SAS-He instrument. The overall design is composed of an optical collector (Fig. 1a) located outdoors connected to a pair of chilled spectrometers (Fig. 1b) and data collections system located indoors within a climate-controlled environment (Fig. 1c). Photons incident on the hemispheric diffuser at the fore-optics of the light collector travel through a large single core optical fiber to a 50/50 bifurcated Y-fiber that diverts the signal to the UV-VIS-NIR and SWIR spectrometers. Within each spectrometer, the light is spectrally dispersed by a diffraction grating and focused onto a

solid-state linear detector array. The array is then read by an electronic interface that passes the data to the computer where it is stored. The data acquisition electronics and spectrometers include an in-line fiber optic shutter for automatic dark signal correction. Dark signals are obtained periodically by closing the in-line shutter and collecting spectra with the same integration time that was used to measure the sky intensity. The thermostatically-controlled (±1°F) refrigerator (Fig.1b) is supplied with dry air and/or desiccant to prevent condensation. The connection of the sky collection optics, rack-mounted data acquisition

equipment and fiber-coupled UV-VIS-NIR and SWIR spectrometers is provided by fiber optic and electrical cables (Fig.1c).



The optical collector, based strongly on the designs of the MFRSR and RSS, includes a hemispheric diffuser and a moving shadow band (Fig. 1a) for distinguishing direct solar and diffuse sky irradiance at approximately 30 sec temporal resolution. A fiber optic umbilical with an in-line shutter connects the optical collector to a pair of commercial off-the-shelf Avantes spectrometers (Fig. 1b) with a wide spectral coverage and high spectral resolution (Table 1). Certainly, this important capability

is superior to the MFRSR, since it increases substantially the opportunity to extend the existing MFRSR-based retrievals to expected multidisciplinary studies with focus on the climate-important properties (Riihimaki et al., 2021). For example, unique absorption and scattering properties of ice and liquid water cloud particles can be retrieved from cloud-transmitted radiance spectra measured with a wide spectral coverage and high spectral resolution (e.g., LeBlanc et al., 2015). The main consideration in the design of the SAS-He instrument was to obtain high radiometric repeatability and efficiency. To address this challenge,

several modifications of the MFRSR-like configuration have been made. The next sections highlight these valuable modifications.

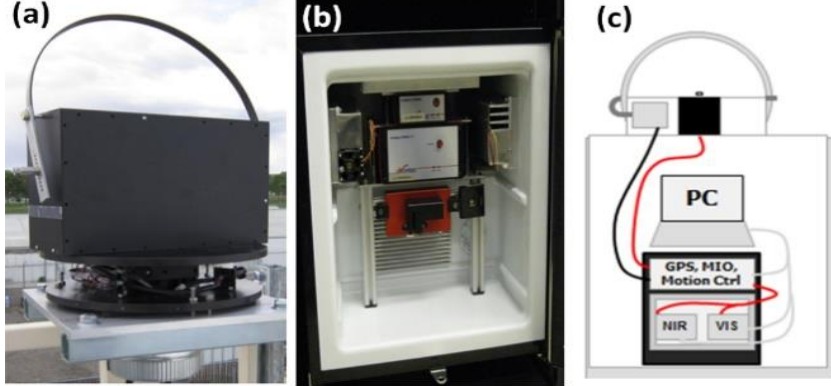

**Figure 1: The SAS-He design: (a) shadow band and sky collection optics mounted outside, (b) UV-VIS-NIR and SWIR**
**spectrometers housed inside the chiller, (c) instrument layout concept.**

**Table 1. A pair of SAS-He spectrometers: Main characteristics**

| Spectrometers | Range | Resolution |
|---|---|---|
| UV-VIS-NIR fiber-coupled spectrometer (Avaspec ULS 2048 CCD) | 350-1040 nm | ~2.4 nm |
| SWIR fiber-coupled spectrometer (Avaspec NIR256-1.7) | 990-1700 nm | ~ 6 nm |
| Angular sampling | Range | Field-of-view |
| Total and diffuse irradiances | Full hemisphere | ~ 3.6 deg (shadowband full-angle) |



## 2.2 SAS-He calibrations

Several aspects of the SAS-He require careful characterization and/or calibration. Some of these are commonly applied to legacy solar radiometers, but some are unique to the hyperspectral measurements of the SAS-He. These corrections and calibrations are described briefly below.

### 2.2.1 Spectral registration

The Avantes grating array spectrometers, calibrated in terms of wavelength prior to delivery, include a multi-order polynomial fit providing wavelength as a function of pixel index number. We confirmed the wavelength mapping is accurate to within a pixel by confirming the location of known emission lines from a Mercury-Argon discharge lamp. In addition, reference to sharp Fraunhofer lines (emission and absorption lines in the solar spectrum) in the UV and visible spectra and well-known atmospheric absorption features including water vapor bands and the oxygen A-band permit in-field confirmation of the spectral registration, a practice which is not typically feasible for filter-based measurements.

### 2.2.2 Spectral resolution

In addition to the pixel-to-wavelength mapping, Avantes also provides the approximate spectral resolution for each spectrometer configuration. For the Si CCD spectrometer, the nominal spectral resolution is about 2.5 nm full width at half maximum (FWHM). The nominal spectral resolution of the InGaAS array spectrometer is about 6 nm FWHM. Note that the spectral resolution is distinct from the pixel spacing which is the wavelength difference between adjacent pixels as inferred from the spectral registration above. The pixel spacing for the Si CCD is about 0.55 nm. The pixel spacing for the InGaAs array is about 3.5 nm. This means that the spectra from the Si CCD is being over-sampled by about a factor of four, while the InGaAs array is being over-sampled by about a factor of two.

### 2.2.3 Internal stray light within the spectrometer

Stray light in the spectrometer, similar to out-of-band leakage for narrow-band filter measurements, represents signal from other wavelengths detected and ascribed to the intended wavelength. We have measured stray light scattered internally within the spectrometer by scanning a double-slit monochromator positioned in front of a broadband light source over the spectral range of the spectrometer. Except for a few isolated "hot pixels," the stray-light levels are below 0.1 to 0.01% over most of the spectral range as shown in Figure 2a. The horizontal axis is the wavelength reported by the grating spectrometer. The vertical axis is the source wavelength provided by the scanning monochromator. The vivid diagonal line indicates that most light is detected at the spectrometer pixels corresponding to the monochromator selected wavelength. However, some spectrometer artifacts are apparent as whisps about the diagonal line, and a ghostlike diagonal feature offset from the diagonal by about 200 nm. The color scale is log-base 10, so a value of -3 (cyan) is 0.1% intensity of stray light relative to the peak signal intensity. Although initially assumed to represent a negligible contribution, the accumulation of even these low





scattering levels generated aberrant behavior at the wavelength limits of the UV/VIS spectrometer that required empirical

correction.

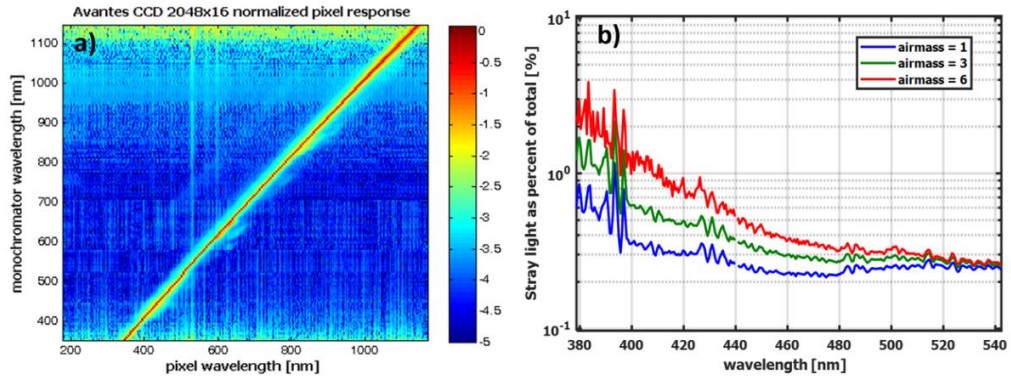

.

**Figure 2: Panel (a) shows results of a stray light measurement for the SAS-He Si CCD obtained by placing a scanning monochromator at the entrance port of the Si CCD grating spectrometer. The horizontal axis is the wavelength**
**reported by the grating spectrometer. The vertical axis is the source wavelength provided by the scanning monochromator. Panel (b) shows the empirical stray light correction of the SAS-He direct horizontal irradiance as a proportion of the raw uncorrected signal for three airmass values over the short wavelength range where this correction is most significant.**

Figure 2b illustrates that the stray light correction is more significant at shorter wavelengths and depends strongly on airmass.

The result of the stray light is that the Langley calibrations for short wavelengths become skewed and artificially shallow,

leading to low calibration bias and low retrieved $AODs$. The empirical correction shown in Figure 2b yields satisfactory

calibration up to 380 nm.

**2.2.4 External stray light detected by the spectrometer**

There are two potential sources of external light affecting the SAS-He irradiance measurements. The first is direct sunlight

leaking through the fiber-optic jacketing of the umbilical. By exposing and shading the optical collector under direct solar

exposure we have confirmed that leakage, if it exists at all, is at undetectable levels. In addition, since 2019 the in-line shutter

(the red device in Fig. 1b) has been moved out of the chiller and incorporated into the collector head such that fiber leakage

would represent common signal and be subtracted as "dark" counts.

165        The second potential source of stray light is from reflective objects near the collector head producing glint detectable by

the SAS-He in the diffuse hemispheric irradiance measurement. This was observed and documented for the Go-Amazon

deployment (not part of this study) due to the proximity of the SAS-He adjacent to a 10-meter-high stainless-steel aerosol

sampling stack. This unfortunate configuration was avoided for the deployments in this study.





### 2.2.5 Non-Lambertian response of the optical collector, "cosine correction"

Hemispheric collectors used by the MFRSR and the SAS-He exhibit a dependence on the angle of incidence of incoming light. For an ideal Lambertian diffuser this dependence is exactly the cosine of the incident angle. The angular response of the SAS-He collector is carefully characterized in lab measurements where the collector is mounted on a rotating stage and exposed to a stationary light source. The lab-derived "cosine correction" is applied to the direct beam measurements, and a variant of the cosine correction is also applied to the diffuse field as the average of the modeled response to isotropic, clear-sky, and overcast
conditions.

### 2.2.6 Spectrometer signal non-linearity

Photodiodes used in the MFRSR and CSPHOT instruments have demonstrated excellent signal linearity spanning several orders of magnitude. In contrast, the SAS-He spectrometers require careful linearity characterization. By varying incident light levels and integration times, we have documented the non-linearity for each grating spectrometer. The non-linearity is small,
though not negligible. To first order, the non-linearity of the direct irradiance measurement becomes incorporated in the cosine correction described above, but the diffuse hemispheric component requires further correction. To infer this correction, we apply the following two-step approach. First, we calculate the direct-to-diffuse ratio by dividing the direct-normal irradiance by diffuse hemispheric irradiance at a given wavelength. We use the direct-normal irradiances and diffuse hemispheric irradiances measured by two collocated ground-based instruments, namely the SAS-He and MFRSR. It should be emphasized
that the calculated SAS-He and MFRSR direct-to-diffuse ratios are calibration-independent in the sense that this is a unitless ratio. Second, we obtain an empirical non-linearity correction to the SAS-He diffuse hemispheric irradiance by dividing the calculated SAS-He ratio by its MFRSR counterpart and then applying the second order polynomial fit (Fig. 3). We apply this correction to the SAS-He diffuse hemispheric irradiance. Its corrected values are used for the corresponding assessment of the SAS-He products (Sect. 4).

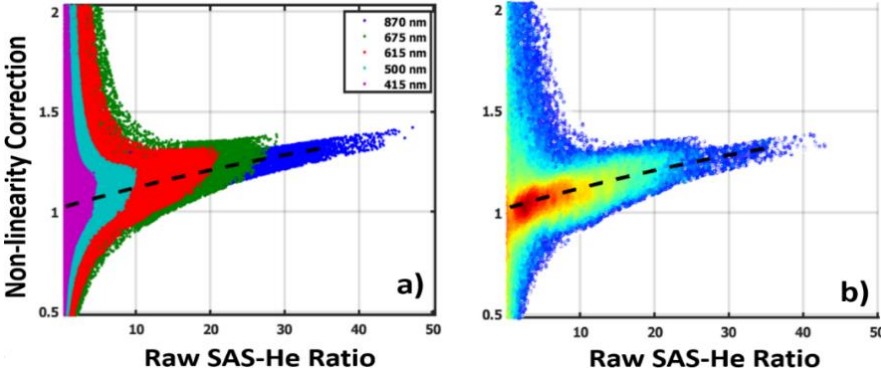


**Figure 3: Example of non-linearity correction as a function of the SAS-He direct-to-diffuse ratio obtained during the TRACER. (a) Relationship displayed for five MFRSR filter wavelengths (415, 500, 615, 675, 870nm) shows a progression along a common tendency. (b) A density plot of all corrections irrespective of filter wavelength. Dashed lines indicate the polynomial fits (a,b).**





### 2.2.7 Spectrometer signal temperature sensitivity

The CCD spectrometers have been confirmed to exhibit a temperature response of less than 0.1% per degree. The InGaAs spectrometers show higher temperature dependent sensitivity, but this is mostly due to changes in thermal background levels that we address through frequent (~30 sec interval) dark measurements. We have also identified that the InGaAs spectrometers show a trough in their temperature response, so we operate our chiller centered on this minimum in temperature sensitivity.

### 2.2.8 Spectrometer responsivity

A Newport Oriel OPS-Q250 and 200 W QTH lamp with NIST-traceable spectral calibration from 250-2400 nm was used to measure the SAS-He relative spectral response. Based on reproducibility, the absolute uncertainty of the measurement is estimated to be up to several percent, insufficient for independent irradiance calibration. However, comparison of normalized responsivity curves shows relative variation less than 0.1% over the full spectral range after a 10-min lamp settling time. The relative spectral responsivity is scaled to agree with Langley calibration to top-of-atmosphere solar irradiance (Fig. 4) using the conventional approach (Kindel et al., 2001; Michalsky and Kiedron, 2022).

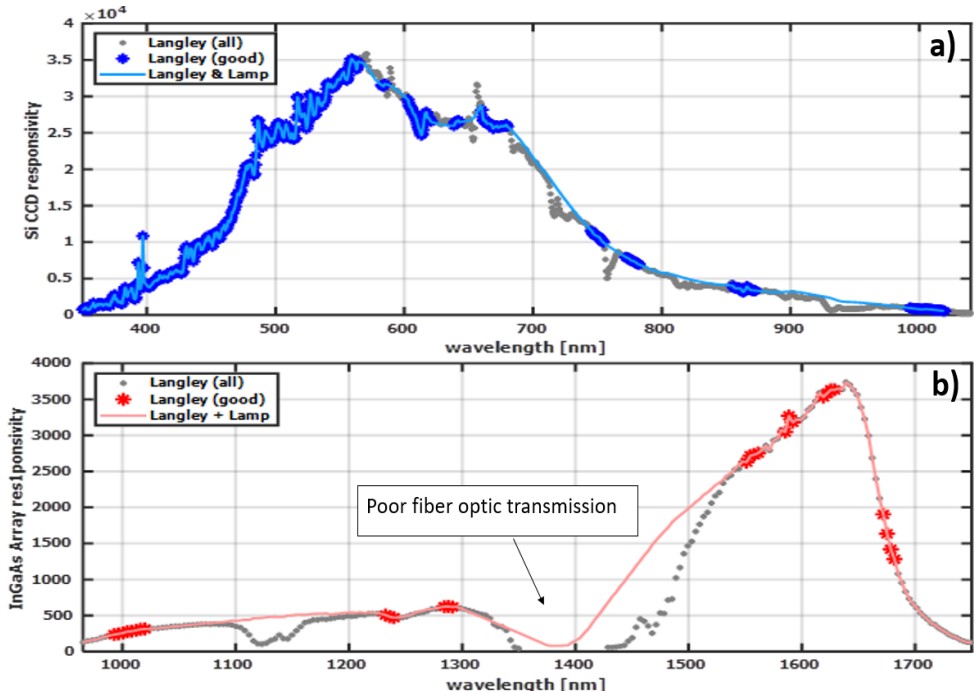

**Figure 4: The spectral responsivity of the Si CCD (a) and InGaAs (b) array. In both figures (a,b) the gray dots show the spectral responsivity obtained through dividing the Langley regression Y-intercept by top-of-atmosphere solar irradiance at each wavelength. The colored asterisks represent those wavelengths where Langley regressions are anticipated to be dependable under suitable atmospheric conditions. The fine blue (Si CCD) and pink (InGaAs Array) lines represent the responsivity obtained as a combination of Langley calibration with lamp calibration. These lines are labeled as "Langley + Lamp."**





### 2.2.9 Langley calibration (direct irradiance)

To determine $AODs$ from the SAS-He measurements, in-field calibration with Langley regressions is applied. Recall that the Langley regressions are linear regressions of log of the measured irradiance versus airmass and they are computed on a twice daily basis. The output of these regressions (Fig. 5) is used for field calibration of the SAS-He. Since the daily Langley regressions exhibit significant noise mostly due to atmospheric variability, several weeks of SAS-He operational measurements are required to accumulate enough acceptable Langley regressions with small (below 1% per day) statistical variability. Application of a stable daily calibration to the SAS-He radiometric measurements allows one to calculate time series of $TOD$ for each wavelength.

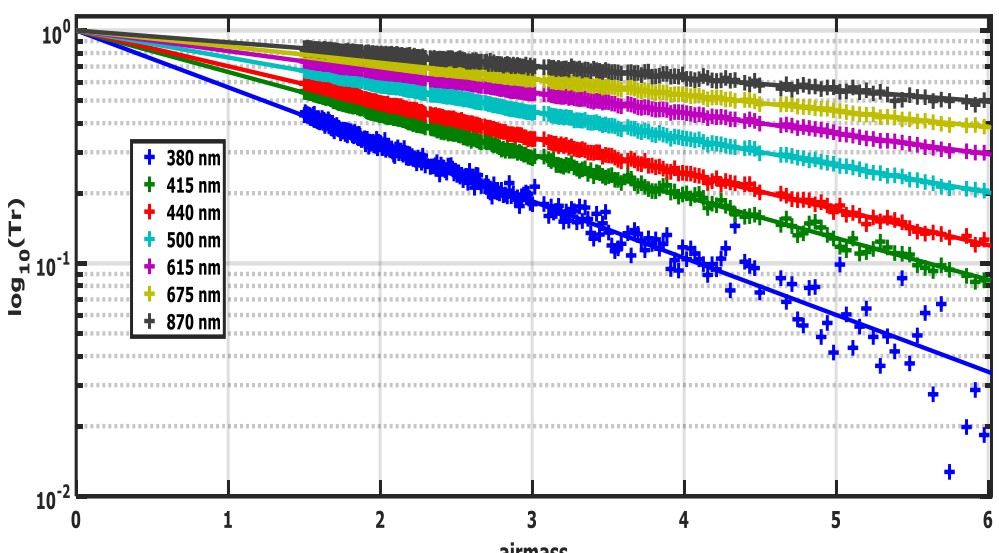

**Figure 5: Example of a SAS-He Langley plot from July 18, 2021 at several CSPHOT and MFRSR wavelengths during TRACER. To demonstrate the expected monotonic behavior with wavelength (with shorter having the steepest slopes and largest optical depths) more clearly, the signal for each wavelength is normalized against the Y-intercept of the Langley regression line forcing each to intersect at 0 airmass.**

### 2.2.10 Total optical depth and gaseous absorption

Figure 6 shows an example of the $TOD$ spectra obtained from the SAS-He "UV-VIS-NIR" spectrometer. The first "UV/VIS" segment (from about 400 to 450 nm) does not include gaseous absorption. The exception is very small $NO_2$ absorption, which can be neglected for many practical applications. Thus, the $TOD$ spectrum in this spectral range represents sufficiently well the actual $AOD$. The second segment (from about 450 to 750 nm) includes substantial ozone absorption in the Chappuis band. The finding of the ozone optical depth requires an estimate of the columnar amount of ozone. Data from the Total Ozone Mapping Spectrometer (TOMS; http://toms.gsfc.nasa.gov) or from the Ozone Monitoring Instrument (OMI;





http://aura.gsfc.nasa.gov/instruments/omi.html) can be used for obtaining the required ozone amount for a given SAS-He location. We continue with spectra obtained from the SAS-He "SWIR" spectrometer (Fig. 6). The gaseous absorption is quite large for the majority of spectral regions, and thus it precludes the straightforward inference of $AOD$ from the measured $TOD$ spectrum. There are, however, several spectral areas with minimal gaseous absorption (e.g., segments around 1020 and 240    1620 nm wavelengths) where the gaseous absorption can be accounted for.

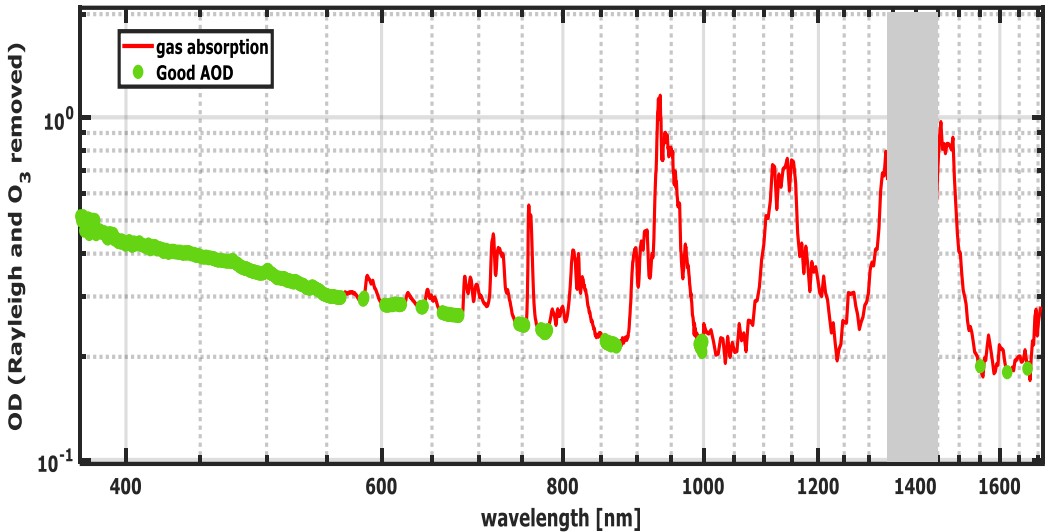

**Figure 6: Example of $TOD$ spectrum with the Rayleigh component and O₃ removed during the TRACER. The green symbols show approximate spectral regions where the $AOD$ may be effectively retrieved. $AODs$ may be found across short wavelength segments from about 400 to 675 nm and about 778 and 870 nm. Near-infrared segments suitable for** 245    **retrieving of $AOD$ are centered on 1020 nm and 1623 nm wavelengths, but care must be used to avoid regions of strong gaseous absorption. The SAS-He fiber optics are opaque near 1400 nm thus this region has been greyed out.**

## 3 ARM-supported campaigns

The evaluation of a new instrument requires two main considerations. The first consideration is a wide range of observational conditions associated with different aerosol and cloud types and strong variability of aerosol loading. The second consideration 250    is availability of good-quality data offered by collocated and coincident instruments with different designs and operations. The three ARM Mobile Facility (AMF) campaigns selected for this analysis provide ground-based instruments for measuring aerosol, cloud, precipitation, and atmospheric state properties and satisfy these challenging considerations. The interested reader can find detailed descriptions of these campaigns at www.arm.gov, which are of significant scientific interest, their suites of ground-based instruments, with state-of-the-art capabilities, and the corresponding data (Berg et al., 2016; Jensen et 255    al., 2022; Russell et al., 2021). Here we summarize these campaigns conducted over climatologically important regions and highlight only data used for our evaluation, specifically, the spectrally resolved $AOD$, $TOD$, as well as the diffuse irradiance and $DDR$.





For each campaign, we compare data provided by MFRSR, CSPHOT and SAS-He. The quality assured (level 2) *AODs* measured at seven (380, 440, 500, 675, 870, 1020, 1640 nm) wavelengths by CSPHOT with a sun-pointing design are used as
a "reference" for evaluation of SAS-He *AODs* (Sect. 4.1). The spectrally resolved *DDRs* and diffuse irradiances provided by MFRSR are used as a "reference" for evaluation of their SAS-He counterparts (Sect. 4.2, 4.3). Recall, MFRSR and SAS-He are sensors with hemispherical receptors that are periodically shaded by rotating bands. The evaluation involves two groups of the *DDR*s and diffuse irradiances offered by the MFRSR measurements. The first group defines the MFRSR products obtained at the *five available* wavelengths (415, 500, 615, 675, 870 nm) during the TCAP only. The second group defines the
MFRSR products obtained at the *six available* wavelengths (415, 500, 615, 675, 870, 1625 nm) during the TRACER and EPCAPE.

### 3.1 TCAP

The main objective of the TCAP was to examine the evolution of optical and microphysical properties of atmospheric aerosol transported from North America to the Atlantic and their impact on the radiation energy budget (Berg et al., 2016). To achieve
this goal, the AMF site (42.03°N; 70.05°W) has been deployed on Cape Cod from July, 2012 through June, 2013. Cape Cod is an arm-shaped peninsula situated on the easternmost portion of Massachusetts, along the U.S. East Coast and is generally downwind of several metropolitan areas. The AMF site was located nearby off-shore Martha's Vineyard Coastal Observatory (41.30°N, 70.55°W) with CSPHOT (https://aeronet.gsfc.nasa.gov).

### 3.2 TRACER

The main objective of the TRACER was to examine aerosol-cloud interactions in deep convection over the Houston metropolitan area of Texas (Jensen et al., 2022). To achieve this goal, the AMF site (29.67°N; 95.06°W) has been deployed near La Porte, Texas airport from October, 2021 through September, 2022. This area with frequent isolated convective systems is characterized by distinct aerosol types originated, for example, from urban, industrial, and marine sources. The CSPHOT was collocated with the MFRSR and SAS-He as part of the AMF.

### 3.3 EPCAPE

The main objective of the EPCAPE is to characterize the radiative properties, aerosol interactions, precipitation characteristics, and extent of stratocumulus clouds in the Eastern Pacific across all four seasons (Russell et al., 2021). To achieve this goal, a 12-month deployment of the AMF site (32.87°N; 117.26°W) on Scripps Pier (La Jolla, California) was started on February, 2023. This area with coastal orography and frequently observed transitions from overcast cloud layers to broken clouds is
influenced by distinct aerosol types originated from the Los Angeles-Long Beach urban port megacity. The CSPHOT is collocated with the MFRSR and SAS-He as part of the AMF.





## 4 Evaluation

This section contains comparison of the aerosol-related (both $AOD$ and $TOD$) and radiative (both diffuse irradiance and $DDR$) properties provided by the SAS-He with those offered by the collocated ground-based instruments, specifically the MFRSR

and CSPHOT, during three selected campaigns highlighted above (Sect. 3). We start with assessment of $AOD$ and $TOD$ measured at different wavelengths by the SAS-He and CSPHOT (Sect. 4.1). Then, evaluation of spectrally resolved $DDR$ offered by the SAS-He and MFRSR is presented (Sect. 4.2). Finally, the diffuse irradiances measured at different wavelengths by the SAS-He and MFRSR are contrasted (Sect. 4.3).

### 4.1 AOD: SAS-He versus CSPHOT

When a straight line between the Sun and the ground-based instrument is cloud-free, one can calculate $AOD$ from $TOD$ at a given wavelength (e.g., Giles et al., 2019):

$$AOD(\lambda) = TOD(\lambda) - \tau_{Ray}(\lambda) - \tau_{gas}(\lambda) \,, \tag{1}$$

where $\tau_{Ray}(\lambda)$ is Rayleigh optical depth due to molecular scattering and $\tau_{gas}(\lambda)$ is optical depth due to absorption of atmospheric trace gases, such carbon dioxide, methane, and water vapor. The gaseous absorption is relatively small in

comparison to $AOD$ at five wavelengths (380, 440, 500, 675, 870 nm). In contrast, gas absorption is comparable with $AOD$ at other two wavelengths (1020, 1640 nm). Thus, the corresponding corrections of the gas absorption are required for $AOD$ calculations at these wavelengths (1020, 1640 nm). The implementation of the required corrections to the SAS-He $AOD$ is underway. Here, we compare the available total optical depth adjusted to the Rayleigh scattering, namely $TOD(\lambda) - \tau_{Ray}(\lambda)$, at these wavelengths (1020, 1640 nm). The same $\tau_{Ray}(\lambda)$ is used for such adjustment. The collocated SAS-He and CSPHOT

measurements provide the needed $TOD(\lambda)$. Below, both scatterplots (Fig. 7) and the corresponding main statistics (Tables 2-4) illustrate level of agreement between the SAS-He and CSPHOT products (both $AOD$ and $TOD$).

Substantial changes of aerosol loading are observed during the TCAP (Fig. 7, top). For example, $AODs$ measured by both SAS-He and CSPHOT at 500 nm wavelength can vary over a wide range (roughly from 0.05 to 0.5). The proximity of Cape Cod to the major urban and industrial sources and its frequently downwind location is mainly responsible for the observed

substantial changes of aerosol loading. The majority of points are packed around the 1:1 line (Fig. 7, top). The corresponding slopes are close to one (about 0.95), absolute values of intercept are small (about 0.01 or less), and root-mean-square errors (RMSEs) are within the expected measurement uncertainty of $AOD$ (0.01-0.02) (Table 2). Both scatterplots and main statistics indicate a strong agreement between SAS-He and CSPHOT $AODs$ at the wavelengths considered here. The measurements at longer wavelengths (1020, 1640 nm) show increased scatter (Fig. 7, top) mostly due to reduced signal-to-noise ratio at these

wavelengths. For example, mean value of $TOD$ measured by CSPHOT at 1640 nm wavelength (0.033) is about three times smaller than the mean value of $AOD$ measured by CSPHOT at 500 nm wavelength (0.103). As a result, agreement for $TODs$ at longer wavelengths (1020, 1640 nm) is slightly weaker than that for $AODs$ at shorter wavelengths (380, 440, 500, 675, 870





nm). To illustrate, the smaller value of slope (0.863) is obtained for $TOD$ (SAS-He vs. CSPHOT) at 1640 nm wavelength in comparison with that (0.96) acquired for $AOD$ (SAS-He vs. CSPHOT) at 500 nm wavelength (Table 2).


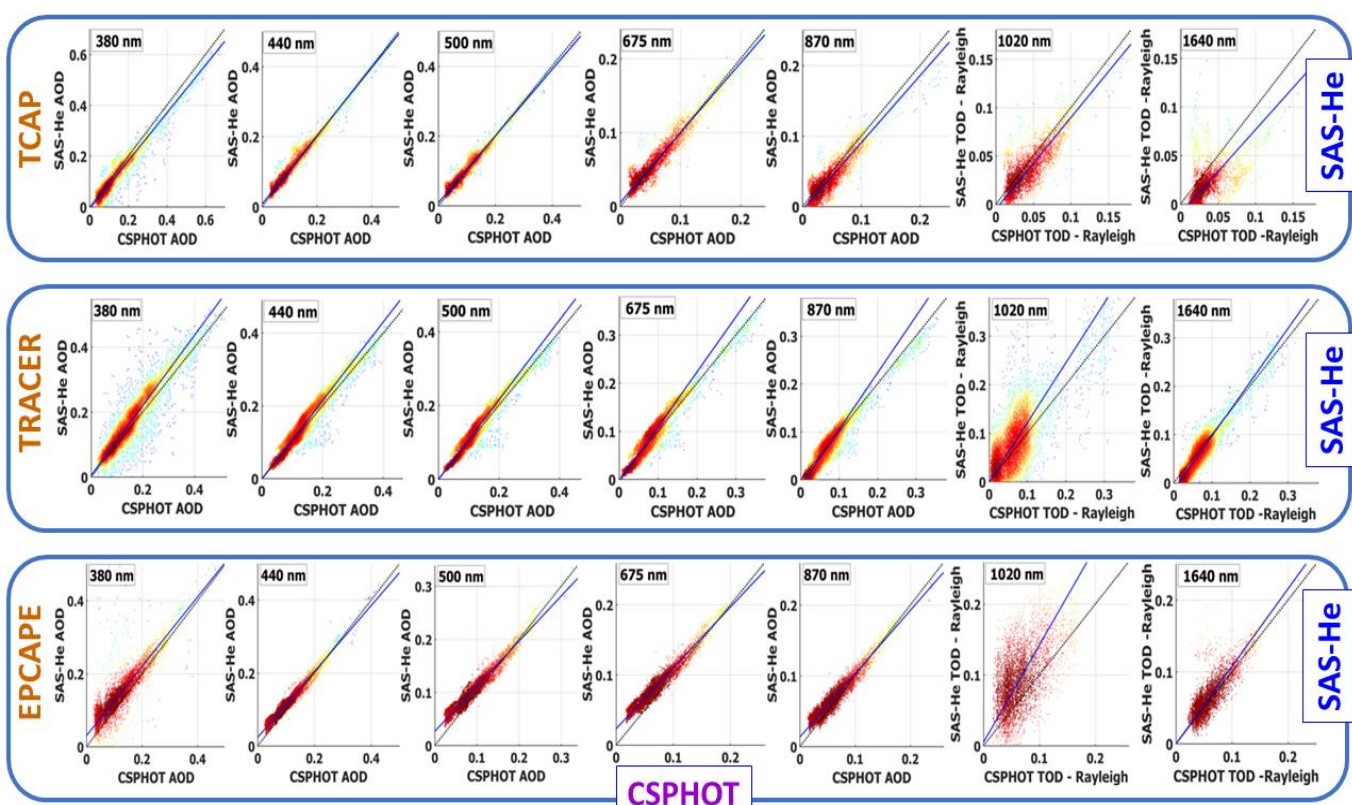

**Figure 7: Density plots of SAS-He $AOD$ versus CSPHOT $AOD$ measured at five (380, 440, 500, 675, 870 nm; first five columns) wavelengths during the TCAP (top row), TRACER (middle row) and EPCAPE (bottom row), respectively. The corresponding scatterplots of $TOD$ adjusted to the Rayleigh scattering at two (1020, 1640nm) wavelengths are also**
**included (last two columns). The short-dashed black line is the 1:1 correspondence line and the solid blue line is the linear regression. Points with light color represent outliers. Note that scales for x- and y-axes depends on wavelength. Tables 2-4 provide the basic statistics of the comparison.**

Similar to the TCAP, the comparisons during the TRACER and EPCAPE show substantial changes of aerosol loading (Fig. 7).
To take an example, $AODs$ measured by both SAS-He and CSPHOT at 500 nm wavelength during the TRACER can vary over a wide range (roughly from 0.05 to 0.4) (Fig. 7, middle). In contrast to the TCAP, the TRACER and EPCAPE show slightly different relationships between SAS-He and CSPHOT $AODs$ in terms of slope. The slopes calculated for the SAS-He and CSPHOT $AODs$ at different wavelengths (380-870 nm) are either larger (1.08-1.18) or smaller (0.85-0.95) than 1.0 for TRACER and EPCAPE, respectively (Tables 3,4). In other words, SAS-He $AODs$ tend to be slightly larger than CSPHOT
$AODs$ during TRACER (Fig. 7, middle). The opposite is true for the EPCAPE (Fig. 7, bottom). The highlighted overestimation and underestimation are likely associated with slightly different SAS-He calibrations performed during the TRACER and





EPCAPE. It appears that these trends have only a minor impact on the corresponding RMSEs. The latter are mostly within the expected measurement uncertainty of $AOD$ (0.01-0.02) despite these trends (Tables 3,4). Also, comparable RMSEs (0.012-0.013) are obtained for $TOD$ at 1640 nm wavelength (Tables 3,4).

**Table 2. Parameters of linear regressions (Fig. 7) obtained for CSPHOT and SAS-He $AOD$s measured at seven wavelengths during the TCAP. Root-mean-square error (RMSE) and number of points (N) are also included.**

|  | 380 nm | 440 nm | 500 nm | 675 nm | 870 nm | 1020 nm | 1640 nm |
|---|---|---|---|---|---|---|---|
| Slope | 0.941 | 0.97 | 0.96 | 0.945 | 0.95 | 0.948 | 0.863 |
| Intercept | -0.007 | 0.008 | 0.009 | 0.005 | -0.005 | -0.005 | -0.01 |
| Bias (y-x) | -0.02 | 0.004 | 0.004 | 0.002 | -0.007 | -0.007 | -0.01 |
| Mean (x) | 0.144 | 0.12 | 0.103 | 0.064 | 0.045 | 0.043 | 0.033 |
| Mean (y) | 0.129 | 0.124 | 0.105 | 0.066 | 0.038 | 0.035 | 0.019 |
| RMSE | 0.016 | 0.008 | 0.008 | 0.007 | 0.01 | 0.01 | 0.007 |
| N | 4213 | 4298 | 4222 | 4328 | 4481 | 4031 | 3220 |

**Table 3. The same as Table 2 except for the TRACER.**

|  | 380 nm | 440 nm | 500 nm | 675 nm | 870 nm | 1020 nm | 1640 nm |
|---|---|---|---|---|---|---|---|
| Slope | 1.08 | 1.08 | 1.08 | 1.15 | 1.18 | 1.23 | 1.12 |
| Intercept | 0.004 | -0.003 | -0.002 | -0.006 | -0.014 | -0.014 | -0.016 |
| Bias (y-x) | 0.02 | 0.008 | 0.007 | 0.004 | -0.003 | 0.01 | -0.008 |
| Mean (x) | 0.153 | 0.127 | 0.108 | 0.072 | 0.059 | 0.063 | 0.064 |
| Mean (y) | 0.169 | 0.135 | 0.115 | 0.077 | 0.055 | 0.077 | 0.055 |
| RMSE | 0.021 | 0.015 | 0.013 | 0.012 | 0.011 | 0.034 | 0.013 |
| N | 7804 | 8503 | 8435 | 8219 | 8058 | 8152 | 7954 |

**Table 4. The same as Table 2 except for the EPCAPE.**

|  | 380 nm | 440 nm | 500 nm | 675 nm | 870 nm | 1020 nm | 1640 nm |
|---|---|---|---|---|---|---|---|
| Slope | 0.95 | 0.904 | 0.85 | 0.867 | 0.895 | 1.37 | 1.08 |
| Intercept | 0.03 | 0.023 | 0.027 | 0.023 | 0.013 | 0.005 | 0.0 |
| Bias (y-x) | 0.02 | 0.01 | 0.01 | 0.01 | 0.007 | 0.03 | 0.004 |
| Mean (x) | 0.116 | 0.099 | 0.087 | 0.068 | 0.059 | 0.062 | 0.057 |
| Mean (y) | 0.14 | 0.113 | 0.101 | 0.082 | 0.066 | 0.091 | 0.061 |
| RMSE | 0.017 | 0.01 | 0.008 | 0.006 | 0.007 | 0.033 | 0.012 |
| N | 7008 | 7513 | 7411 | 7405 | 7490 | 4478 | 4462 |





## 4.2 DDR: SAS-He versus MFRSR


The *DDR* exhibits even wider range of changes (Fig. 8) than the *AOD* (Fig. 7). These significant changes of *DDR* (about two orders of magnitude) are attributed mainly to two factors. First, the *DDR* comparison (Fig. 8; Tables 5-7) includes both clear- and cloudy-sky conditions where a straight line between the Sun and the ground-based instruments was either cloud-free or blocked by a cloud. Here, the term "cloudy-sky" defines all cloud types observed during the selected campaigns. Typically,

different cloud types have distinct and highly variable cloud properties, such as cloud amount and cloud optical thickness, in time and space. Second, the direct and diffuse irradiances vary differently depending on the plume and/or cloud properties. For instance, the *DDR* is small (close to zero; Fig. 8) during the presence of dense plumes associated with strong air pollution emissions or overcast and optically thick clouds. In this case, the direct irradiance is negligible in comparison with the diffuse irradiance. During clean and clear-sky conditions, the direct irradiance reaches large values, while the diffuse irradiance drops

off. These conditions are characterized by large *DDRs* (Fig. 8).

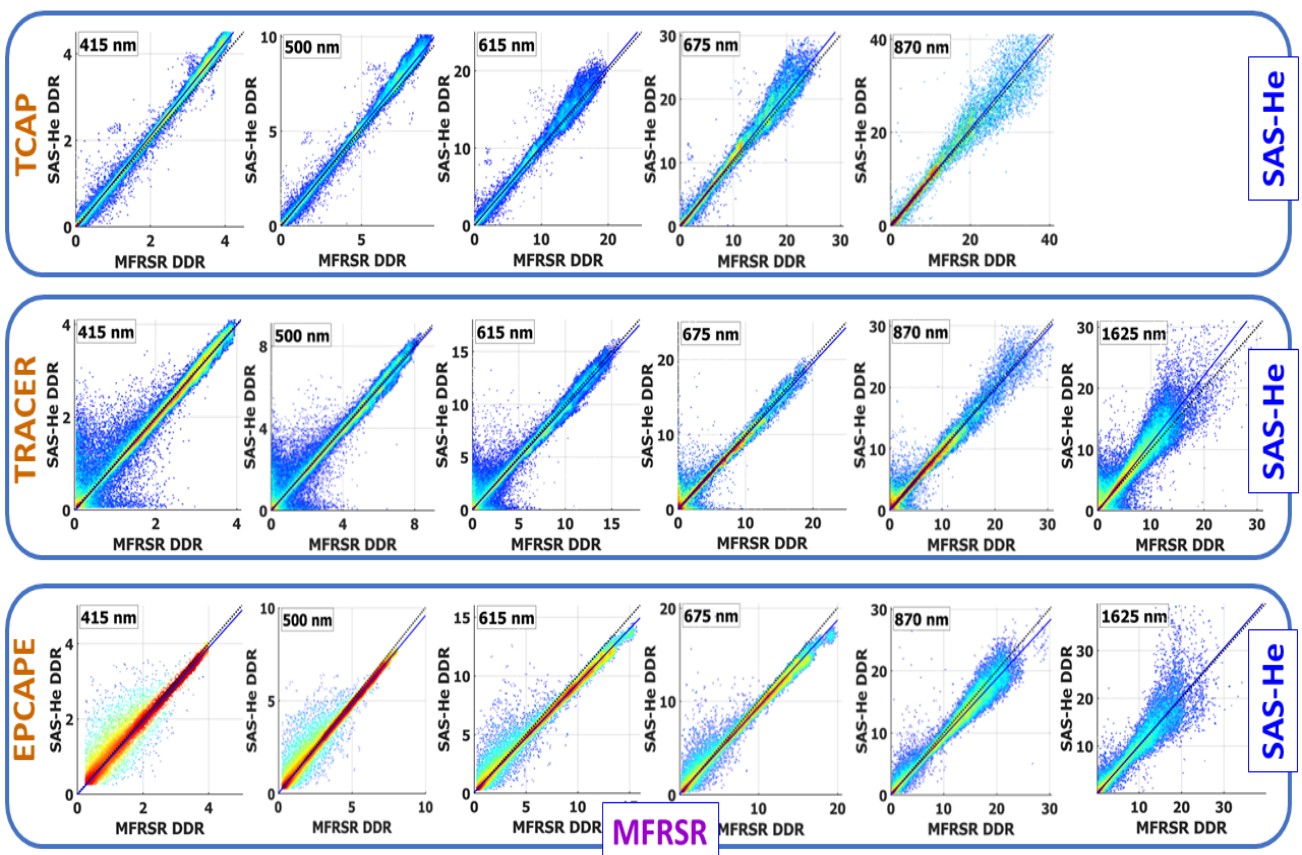

**Figure 8:** **Density plots of SAS-He *DDR* versus MFRSR *DDR* obtained from the direct and diffuse irradiances measured at five (415, 500, 615, 675, 870 nm) wavelengths during the TCAP (top row), and at six (415, 500, 615, 675, 870, 1625 nm) wavelengths during the TRACER (middle row) and EPCAPE (bottom row), respectively. Note that the**

***DDR* is dimensionless.**



**Table 5. Parameters of linear regressions (Fig. 8) obtained at five (415, 500, 615, 675, 870nm) wavelengths for MFRSR and SAS-He _DDR_s during the TCAP. Root-mean-square error (RMSE) and number of points (N) are also included.**

|  | 415 nm | 500 nm | 615 nm | 675 nm | 870 nm |
|---|---|---|---|---|---|
| Slope | 1.08 | 1.05 | 1.03 | 1.05 | 1.04 |
| Intercept | -0.058 | -0.079 | -0.067 | -0.1 | -0.12 |
| Bias (y-x) | 0.09 | 0.1 | 0.05 | 0.2 | 0.07 |
| Mean (x) | 1.88 | 3.28 | 4.1 | 4.88 | 5.06 |
| Mean (y) | 1.97 | 3.37 | 4.15 | 5.03 | 5.12 |
| RMSE | 0.06 | 0.109 | 0.126 | 0.172 | 0.209 |
| N | 62829 | 57178 | 40538 | 37489 | 29199 |

**Table 6. Parameters of linear regressions (Fig. 8) obtained at six (415, 500, 615, 675, 870, 1625nm) wavelengths for MFRSR and SAS-He _DDR_s during the TRACER. Root-mean-square error (RMSE) and number of points (N) are also included.**

|  | 415 nm | 500 nm | 615 nm | 675 nm | 870 nm | 1625 nm |
|---|---|---|---|---|---|---|
| Slope | 1.0 | 0.99 | 0.972 | 0.973 | 0.98 | 1.11 |
| Intercept | -0.039 | -0.035 | -0.003 | 0.001 | -0.027 | -0.18 |
| Bias (y-x) | -0.03 | -0.07 | -0.1 | -0.1 | -0.1 | 0.3 |
| Mean (x) | 2.01 | 3.26 | 4.44 | 4.92 | 5.76 | 4.81 |
| Mean (y) | 1.98 | 3.19 | 4.32 | 4.79 | 5.62 | 5.16 |
| RMSE | 0.072 | 0.119 | 0.18 | 0.208 | 0.29 | 0.52 |
| N | 106443 | 102537 | 96660 | 94278 | 78564 | 61001 |

**Table 7. The same as Table 6 except for the EPCAPE.**

|  | 415 nm | 500 nm | 615 nm | 675 nm | 870 nm | 1625 nm |
|---|---|---|---|---|---|---|
| Slope | 0.979 | 0.958 | 0.93 | 0.931 | 0.935 | 1.02 |
| Intercept | -0.027 | -0.004 | 0.057 | 0.063 | 0.049 | -0.048 |
| Bias (y-x) | -0.08 | -0.2 | -0.4 | -0.5 | -0.5 | 0.04 |
| Mean (x) | 2.44 | 4.27 | 6.51 | 7.43 | 8.5 | 5.08 |
| Mean (y) | 2.36 | 4.08 | 6.11 | 6.98 | 7.99 | 5.12 |
| RMSE | 0.044 | 0.079 | 0.178 | 0.231 | 0.466 | 0.513 |
| N | 36308 | 35993 | 37455 | 36588 | 32087 | 14765 |






A moderate scattering of points with well-defined clustering along 1:1 correspondence line (Fig. 8) indicates that the SAS-He measurements offer the spectrally resolved *DDR* in a reasonable manner. Visually, the scattering of points has campaign-dependent features (Fig. 8). To illustrate, a noticeable number of points are located along the x- and y-axis during the TRACER (Fig. 8, middle). Alternatively stated, the *DDRs* offered by two instruments, namely the SAS-He and MFRSR, occasionally

can be quite different when the *DDR* values are small-to-moderate (less than 3). Combination of several potential reasons, such as cloud-induced variability of both the direct and diffuse irradiances at small scales, a minor temporal mismatch due to different reporting times, and the spatial separation of these neighboring instruments (~120 m), could be responsible for different observational conditions for the two instruments (sunlit vs. shadow cases for two instruments spaced slightly apart), and thus could contribute to the highlighted differences. It appears that the level of agreement between the SAS-He *DDR* and

MFRSR *DDR* (Tables 5-7) is scarcely affected by the campaign-dependent variability of cloud and aerosol properties: the slope is close to one (0.93-1.1), the mean values are comparable, and these values exceed the RMSE substantially (about ten times or more).

## 4.3 Diffuse irradiance: SAS-He versus MFRSR

Scatterplots generated for the diffuse irradiances measured by the SAS-He and MFRSR illustrate clearly that these irradiances,

on average, are in a good agreement (Fig. 9). It should be emphasized that the diffuse irradiances rely on the calibration. Thus, potential calibration-related issues could have a profound impact on the statistical relationship between these irradiances. The MFRSR lamp calibration issue at 1625 nm wavelength is documented for the EPCAPE, and this issue is responsible for the significant disagreement between the diffuse irradiances measured by the SAS-He and MFRSR at 1625 nm wavelength (Fig. 9; bottom). The corresponding slope is very small (0.075) and the difference between mean values of the diffuse irradiances is

enormous (0.942 vs. 0.066) (Tables 8-10). It is also vital to note that the accompanying *DDRs* offered by the SAS-He and MFRSR at 1625 nm wavelength are in very good agreement (Fig. 8; bottom), because the *DDRs*, in contrast to the diffuse irradiances, do not depend on calibration. Similar to the *DDR*-related scatterplots (Fig. 8), the scatterplots generated for the diffuse irradiances (Fig. 9) display the campaign-dependent features of points scattering around the 1:1 correspondence line. For example, the TRACER in comparison with the EPCAPE has a wider spread of points (Fig. 9, middle vs. bottom). The

fraction of points contributing to this spread is small relative to the fraction of points clustering around the 1:1 correspondence line (Fig. 9, middle vs. bottom). Thus, the level of agreement between the main statistics (Tables 8-10) depends slightly on these spread-contributed points.





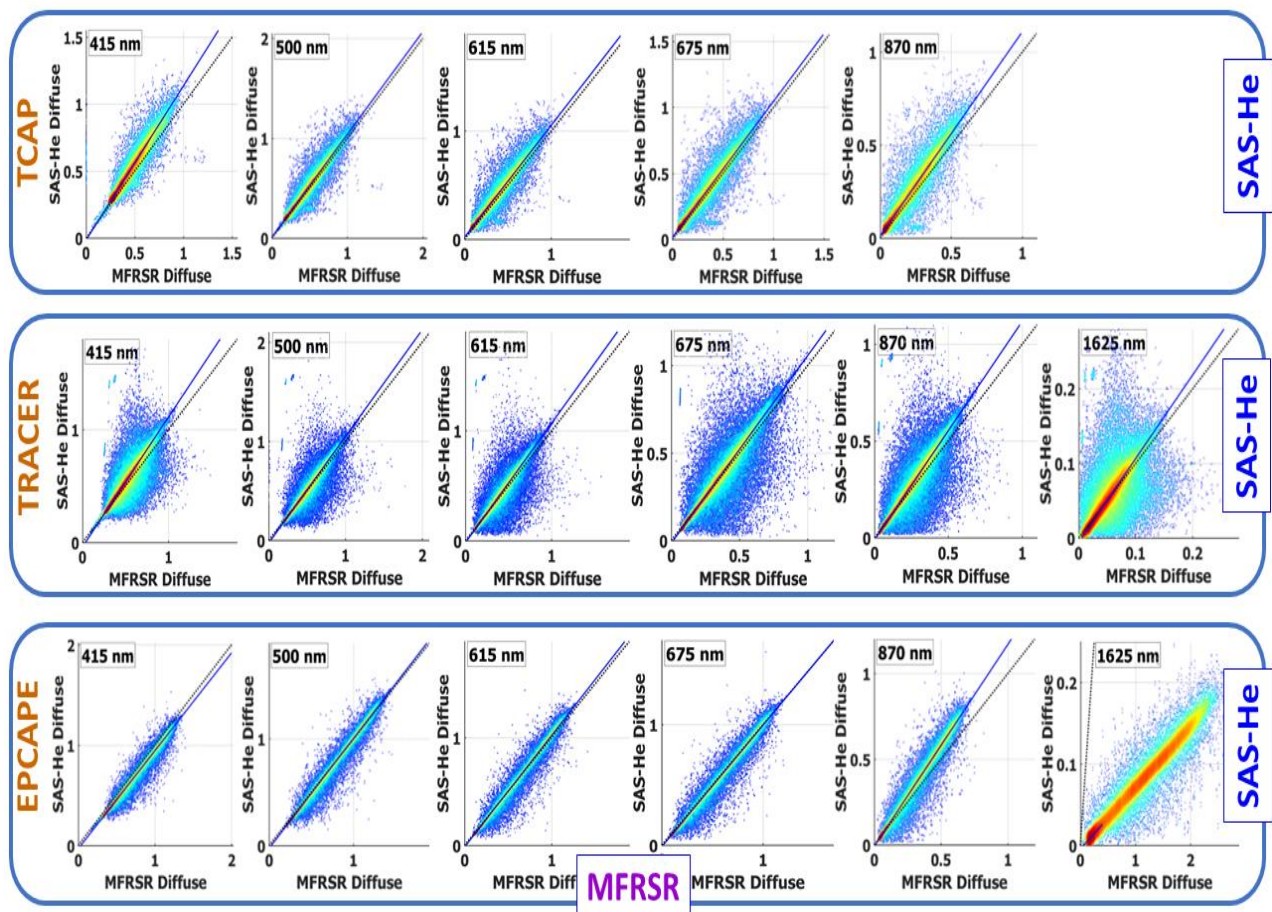

Figure 9: The same as Figure 8 except for the diffuse irradiance (Wm$^{-2}$µm$^{-1}$).






**Table 8. The same as Table 5 except for the diffuse irradiance (Wm⁻²µm⁻¹).**

|          | 415 nm | 500 nm | 615 nm | 675 nm | 870 nm |
|----------|--------|--------|--------|--------|--------|
| Slope    | 1.15   | 1.03   | 1.03   | 1.03   | 1.1    |
| Intercept | -0.016 | 0.011 | 0.019  | 0.017  | 0.012  |
| Bias (y-x) | 0.04 | 0.02  | 0.03   | 0.02   | 0.02   |
| Mean (x) | 0.393  | 0.34   | 0.221  | 0.186  | 0.113  |
| Mean (y) | 0.438  | 0.361  | 0.247  | 0.208  | 0.136  |
| RMSE     | 0.014  | 0.009  | 0.008  | 0.008  | 0.008  |
| N        | 62252  | 55624  | 55053  | 54895  | 58773  |

**Table 9. The same as Table 6 except for the diffuse irradiance (Wm⁻²µm⁻¹).**

|          | 415 nm | 500 nm | 615 nm | 675 nm | 870 nm | 1625 nm |
|----------|--------|--------|--------|--------|--------|---------|
| Slope    | 1.15   | 1.08   | 1.09   | 1.07   | 1.13   | 1.12    |
| Intercept | -0.04 | -0.029 | -0.019 | -0.016 | -0.012 | -0.006  |
| Bias (y-x) | 0.02 | 0.001 | 0.005  | 0.002  | 0.009  | -0.001  |
| Mean (x) | 0.429  | 0.389  | 0.282  | 0.244  | 0.165  | 0.041   |
| Mean (y) | 0.453  | 0.39   | 0.287  | 0.246  | 0.175  | 0.042   |
| RMSE     | 0.014  | 0.013  | 0.011  | 0.01   | 0.009  | 0.004   |
| N        | 122849 | 115156 | 112052 | 111491 | 116957 | 104351  |

**Table 10. The same as Table 7 except for the diffuse irradiance (Wm⁻²µm⁻¹).**

|          | 415 nm | 500 nm | 615 nm | 675 nm | 870 nm | 1625 nm |
|----------|--------|--------|--------|--------|--------|---------|
| Slope    | 0.975  | 1.03   | 1.04   | 1.01   | 1.18   | 0.075   |
| Intercept | -0.033 | -0.028 | -0.016 | -0.014 | -0.01 | -0.005  |
| Bias (y-x) | -0.05 | -0.01 | -0.002 | -0.01  | 0.03   | -0.9    |
| Mean (x) | 0.598  | 0.525  | 0.394  | 0.355  | 0.218  | 0.942   |
| Mean (y) | 0.551  | 0.514  | 0.392  | 0.344  | 0.247  | 0.066   |
| RMSE     | 0.008  | 0.008  | 0.006  | 0.005  | 0.006  | 0.004   |
| N        | 54988  | 51920  | 49595  | 49338  | 54685  | 45482   |



## 5 Summary

We introduce a ground-based radiometer, the so-called Shortwave Array Spectroradiometer-Hemispheric (SAS-He), with an increased spectral coverage (350-1700 nm) and improved spectral resolution. The latter is about 2.4 nm and 6 nm in the UV-VIS-NIR (350-1040 nm) and SWIR (990-1700 nm) spectral ranges, respectively. The SAS-He measures the spectrally resolved total irradiance and its direct and diffuse components with high temporal (30 sec) resolution. Both aerosol optical depth ($AOD$) and total optical depth ($TOD$) are derived from the direct irradiance measured by the SAS-He, while direct-to-diffuse ratio

($DDR$) is calculated using two components of the measured total irradiance. We assess performance of the SAS-He using integrated datasets collected during three field campaigns supported by the U.S. Department of Energy's (DOE's) Atmospheric Radiation Measurement (ARM) Program: (1) Two-Column Aerosol Project (TCAP) (Berg et al., 2016), (2) Tracking Aerosol Convection Interactions Experiment (TRACER) (Jensen et al., 2022), and (3) Eastern Pacific Cloud Aerosol Precipitation Experiment (EPCAPE) (Russell et al., 2021). These campaigns represent climatologically important regions with different

types of aerosols originated from the major marine, urban and industrial sources.

    For our assessment we use data offered by three collocated ground-based instruments, namely Multi-Filter Rotating Shadowband Radiometer (MFRSR), Cimel sunphotometer (CSPHOT) and SAS-He, as part of the ARM Mobile Facility (AMF). Our assessment involves (i) $AOD$ measured at five (380, 440, 500, 675, 870 nm) wavelengths and $TOD$ measured at two wavelengths (1020 and 1640 nm) by the SAS-He and CSPHOT, (ii) the diffuse irradiance and $DDR$ provided by the SAS-

He and MFRSR at five (415, 500, 615, 675, 870 nm) wavelengths during the TCAP, and (iii) the diffuse irradiance and $DDR$ provided by the SAS-He and MFRSR at six (415, 500, 615, 675, 870, 1625 nm) wavelengths during the TRACER and EPCAPE. The measurements of the diffuse irradiance and $DDR$ define all-sky observational conditions when a straight line between the Sun and the ground-based instruments was either cloud-free or blocked by a cloud. Data provided by the CSPHOT and MFRSR are considered as "reference" during our assessment.

We compare the spectrally resolved parameters related to aerosol loading (both $AOD$ and $TOD$) and radiative properties (both diffuse irradiance and $DDR$) supplied by the SAS-He with those provided by the CSPHOT and MFRSR using scatterplots and the main statistics, such as slope and intercept of linear regression, and root-mean-square error (RMSE). Our comparison demonstrates that, on average, the SAS-He properties match closely their MFRSR and CSPHOT counterparts despite the challenging observational conditions associated with large variability of aerosol loading and distinct types of aerosols and

clouds. In particular, the $AOD$- and $TOD$-related RMSEs are within the expected measurement uncertainty of $AOD$ (0.01-0.02) for the majority of cases. Moreover, the slope is mostly close to one (0.85-1.18) and absolute values of intercept are mostly near zero (less than 0.07) for both the aerosol and radiative properties considered here. It is expected that SAS-He data collected for a period exceeding 10 years (since 2011) will be used to derive previously unavailable or enhanced data products of aerosol, clouds, surface (e.g., Riihimaki et al., 2021) at multiple sites with worldwide locations and these ground-based products

combined with those offered by aircraft and satellite observations (Remer et al., 2023) will be imperative in the context of evaluation and improvements of model predictions.



## Data availability

Data can be downloaded from the ARM data archive (https://www.arm.gov/data/).
CSPHOT: http://dx.doi.org/10.5439/1461660
MFRSR: http://dx.doi.org/10.5439/1356805
MFRSR7nch: http://dx.doi.org/10.5439/1756632
SAS-He vis: http://dx.doi.org/10.5439/1110768
SAS-He nir: http://dx.doi.org/10.5439/1110710

## Author contribution

Conceptualization, E.K. and C.J.F. with input from J.C.B. Data processing, B.D.E. Formal Analysis, E.K. and C.J.F. Writing
– Original Draft, E.K.; Writing-Review & Editing, E.K. and C.J.F. with input from J.M.C.

## Competing interest

The author declare that they have no conflict of interest.

## Acknowledgements

This research was supported by the U.S. Department of Energy (DOE), Office of Science Biological and Environmental
Research, as part of the Atmospheric Radiation Measurement (ARM) user facility. The Pacific Northwest National Laboratory
is operated for DOE by the Battelle Memorial Institute under Contract DEAC05-76RL.

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
