# Peer review of "Shortwave Array Spectroradiometer-Hemispheric (SAS-He): Design and Evaluation"

_Atmospheric Measurement Techniques, 2024_

## Author Comment (AC1)

RC1: **'Comment on amt-2024-13'**, Anonymous Referee #1

We highly appreciate your comments on our manuscript. We hope that you will find our responses and the corresponding revisions for the original manuscript satisfactory. Please find below your comments/suggestions (blue color) and our responses (red color).

The paper is suitable for AMT, and I recommend publishing after correcting web links and references as suggested in my technical recommendations below.

Technical suggestions:

44: *infrequently (typically several times a day) and do not capture the diurnal cycle*

There are now Geostationary aerosol measurements from NASA:

https://ladsweb.modaps.eosdis.nasa.gov/missions-and-measurements/applications/geoleo/

Done (lines 44-46)

72  add MFRSR SSA references:

Mok, et al., Comparisons of spectral aerosol absorption in Seoul, South Korea, *Atmos. Meas. Tech.*, https://doi.org/10.5194/amt-11-2295-2018 , 2018

Mok, et al., Impacts of atmospheric brown carbon on surface UV and ozone in the Amazon Basin, *Sci. Rep.*, https://doi.org/10.1038/srep36940 , 2016

Corr, et al., Retrieval of aerosol single scattering albedo at ultraviolet wavelengths at the T1 site during MILAGRO (2009), *Atmos. Chem. Phys.*, 9, 5813–5827, https://doi.org/10.5194/acp-9-5813-2009 , 2009

Krotkov, et al., Aerosol ultraviolet absorption experiment (2002 to 2004), part 2: absorption optical thickness, refractive index, and single scattering albedo, Opt. Eng., 44, 041005, https://doi.org/doi:10.1117/1.1886819  , 2005

Added (lines 74-78)

*143:.* Better to move to Figure 2 caption. Explain color bar units.

Good suggestion!  Moved, and added description of colorbar within caption (lines 162-164).

Section 2.2:

Describe if tilt and misalignment corrections if applied, e.g.,

Mikhail D. Alexandrov, Peter Kiedron, Joseph J. Michalsky, Gary Hodges, Connor J. Flynn, and Andrew A. Lacis, "Optical depth measurements by shadow-band radiometers and their uncertainties," Appl. Opt. **46**, 8027-8038 (2007)

Thank you for the reference. We have not applied the tilt and misalignment corrections (now noted in lines 195-200) due to having been installed in a solid surface as opposed to ground susceptible to shifting.

In 2.2.5 give more details of the angular calibration and figure with the Lab optical setup and example of the measured cosine response at different wavelengths.

Done. New figure (Figure 3) and details have been added regarding the cosine correction (lines 186-193).

Describe corrections to the diffuse irradiance due to blockage of the forward scattered light by shadow-band (aureole correction), e.g.,

*Min, Q., E. Joseph, and M. Duan (2004), Retrievals of thin cloud optical depth from a multifilter rotating shadowband radiometer, J. Geophys. Res., 109, D02201, doi:10.1029/2003JD003964.*

Clarification (lines 83-86) and the corresponding references (Segal-Rosenheimer et al., 2013; Min et al, 2004; Norgren et al., 2022) have been added.

In 2.2.6 give a more detailed description of the non-linearity correction if this has not been published before or give a reference.

Done (lines 211-217).

Acknowledge that MFRSR DDR is biased for coarse aerosols (e.g., dust) and cirrus clouds, due to the blockage of the forward scattered aureole light as discussed in Min et al., JGR 2004

Done (lines 83-86).

170-175: Show lab setup and add figure with the lab measured angular response at different wavelengths.

New figure (Figure 3) shows response at different wavelengths as a function of incident angle.

179:  Add figure which shows results of non-linearity testing or add reference.

Figure 4 shows the results of the non-linearity testing.

180-181: clarify this sentence:  *To first order, the non-linearity of the direct irradiance measurement becomes incorporated in the cosine correction described above*

Clarified (lines 211-217).

Figure 4: Give units for Y-axis

Done, thank you!

231:  Gaseous NO2 absorption becomes important for single scattering albedo retrievals at small AODs, e.g.,  Krotkov, et al  (2005), Partitioning between aerosol and NO2 absorption in the UVA (https://doi.org/10.1117/12.615285)

Highlighted (lines 276-277).

235-236: Give proper references and correct URLs:For TOMS total ozone product:     https://acd-ext.gsfc.nasa.gov/anonftp/toms/

TOMS Science Team (Unreleased), TOMS Nimbus-7 Total Column Ozone Daily L3 Global 1 deg x 1.25 deg Lat/Lon Grid V008, Greenbelt, MD, Goddard Earth Sciences Data and Information Services Center (GES DISC), Accessed:

, https://disc.gsfc.nasa.gov/datacollection/TOMSN7L3dtoz_008.html

Done (lines 279-280).

For OMI total ozone product:

Pawan K. Bhartia (2012), OMI/Aura TOMS-Like Ozone and Radiative Cloud Fraction L3 1 day 0.25 degree x 0.25 degree V3, NASA Goddard Space Flight Center, Goddard Earth Sciences Data and Information Services Center (GES DISC), Accessed , https://doi.org/10.5067/Aura/OMI/DATA3002

For OMI instrument: https://aura.gsfc.nasa.gov/omi.html

Done (lines 280-281).

Figure 7 and Tables 3,4:  Explain significant differences in AOD comparisons at 1020nm during TRACER and EPCAPE.

Thank you for pointing that out! We had noticed this previously. However, we had not been able to identify this spectrometer degradation issue until recently.  We have replaced the previous plots, which used the 1020 nm pixel from the degraded NIR spectrometer with new figures using the corresponding pixel from the UV/VIS spectrometer. Such replacement yields consistent agreement for all three campaigns. We attribute the previous disagreement at the TRACER and EPCAPE to degradation of the NIR spectrometer affecting its short wavelength range, but apparently leaving the 1.6 micron values unaffected.

Figure 8: Extend Y axis to show full range of DDR variability.

Done.

---

## Author Comment (AC2)

RC2**: 'Review of "Shortwave Array Spectroradiometer-Hemispheric (SAS-He): Design and Evaluation"**

We highly appreciate your comments on our manuscript. We hope that you will find our responses and the corresponding revisions for the original manuscript satisfactory. Please find below your comments/suggestions (blue color) and our responses (red color).

This is a great manuscript to read, however there are a few minor comments to address, mostly on some clarification of some points (see list below). After these minor comments are addressed, it is recommended for publication in AMT.

*General Comments:*

1. The direct to diffuse ratio is both used for quantifying the non-linearity of the spectrometers and the comparison to MFRSR. It is not evident if the subset of data is used is the same for non-linearity correction and comparison to MFRSR, and if there is a circularity in these comparisons.

In fact, there is circularity built in, in that we require the direct-diffuse ratio to agree, and we obtain this agreement by applying the non-linearity correction derived from the initial disagreement between the MFRSR and SASHe measurements. Our argument that this approach is valid because it appears to be independent of wavelength and of deployment (both the TRACER and EPCAPE show essentially the same correction).

*Specific Comments:*

1. Line 24-25 (repeated in the summary): How does the uncertainty of the cimel sunphotometer translate to the uncertainty metric from this method 0.01-0.02 root mean square error, may not exactly equate to the accuracy uncertainty. Some refinement in this statement to differentiate uncertainty in accuracy and root mean square.

This statement has been refined (lines 24-25), the corresponding clarifications (lines 312-314, 390-391) and reference (Shinozuka et al., 2013) have been added.

2. Line 60-61: How does the SWS influence the development of the SAS-He? Was it at all influence? If so then a citation might be adequate here.

No real influence from the SWS, but the 4STAR was an influence. The corresponding reference ( Dunagan et al., 2013) has been added.

3. Line 88: if it is a single core fiber, how does the coupling work to split exactly 50/50 in the Y fiber optic?

It is only a nominal 50/50 split, but the two spectrometers are independently calibrated with Langleys so the ratio is almost immaterial.

4. Line 93: Wouldn't a celsius scale be more appropriate to highlight the stability of the thermal control?

The thermal control used allows either +/- 1 C or +/- 1 F.  Using F achieves 5/9 of the resolution of C.

5. Spectral Registration: How often is this procedure run? since 3 campaigns are shown, has this been done only once, or is there a time series of spectral calibration to ensure spectrometer stability?

It has been measured several times, once or twice per campaign, with no changes by more than a pixel and nothing systematic observed over time.

6. Spectral Resolution: How was this measured?

Not measured, vendor-provided nominal resolution for our purchased spectrometer configuration.

7. cosine correction: How far off from cosine was the lab measurements returned?, and is it spectrally neutral? While most of this can be corrected, if substantial, it would indicate that sampling is uneven, so there is increased error in the hemispherical measurements.

We've provided a new figure (Figure 3) showing the cosine correction.  It is spectrally neutral.

8. Spectrometer non-linearity correction: There seems to be the whole range of potential non-linearity correction factors at low direct to diffuse ratios (vertical color range in figure 3a) Does this mean that the correction is limited in efficacity at the lowest end of the ratio? There is no mention that there are limits to the correction factor at low ratios.  This seems potentially problematic to accurate measurements.

We are inclined to believe the spread in the nonlinearity factor toward the low end is driven by statistics and the finite digital resolution of the A/D counters.  Even though the spread appears large, recall that this is a density plot so the correction factor is actually pretty robust and quite reproducible to within a few percent at worst.

9. Line 196: degree Celsius or Fahrenheit?

Fahrenheit

10. Line 198: How much variation is observed in the 30 second dark measurements?

They are quite stable except for occasional partial shutter operation. The processing software identifies and reject spectra where the shutter was not either entirely closed (for darks) or entirely open (for "lights").

11. Line 199: What is the temperature of the minimum in temperature sensitivity?

It was 29 F, but we opt to operate slightly warmer (35F) to reduce the potential of condensation/freezing issues.

12. Figure 5: What is the standard deviation from the fit of the langley calibration presented here? And the resulting impact to the expected accuracy in AOD?

This is a single Langley, not a compilation of Langleys (as is used for the 4STAR). In analog to the MFRSR we apply an interquartile filter to Io values over a span of several weeks, and then apply a sliding Gaussian-weighted averaging filter to determine a daily Io. The daily Io values are stable to within a few tenths of a percent per day, which effects OD at less than 0.01. Absolute accuracy is very difficult to determine which is why we are comparing to collocated measurements.

13. Figure 6: How about the wavelengths near 1220 nm for AOD?

We agree that the wavelengths near 1220 nm should be suitable for the AOD measurements due to small gaseous absorption. However, AODs at these wavelengths are not provided by the reference instruments (both MFRSR and CSPHOT) considered in our paper. Thus, we use only AOD offered by the reference instruments.

14. Line 299: Is 'as' missing for 'such as'?

Done (line 348).

---

## Author Comment (AC3)

**RC3: 'Comment on amt-2024-13', Anonymous Referee #3**

We highly appreciate your comments on our manuscript. We hope that you will find our responses and the corresponding revisions for the original manuscript satisfactory. Please find below your comments/suggestions (blue color) and our responses (red color).

I believe that this manuscript aligns perfectly with the scope of AMT, and the presented results are indeed relevant. There are only a few minor technical remarks to address.

**Minor/technical comments:**

Page 2, line 47: Following Giles et al. (2019), AERONET derives AOD at 9 different spectral bands although the AOD at 935 nm is extrapolated based on the Ångström Exponent. Why did the authors state in the text that AERONET provides information at seven spectral bands?

Corrected (lines 49-50).

4: There is a typo in the y-axis? Are these values unitless?

Fixed (Figure 5).

Section 2.2.8 and Figure 4: What atmospheric conditions are considered "good" by the authors in the Figure? I believe that some explanation about this classification in the text would be necessary. Additionally, are the spectral bands not included (marked with asterisks) due to the influence of atmospheric gas absorption?

Yes.

Have the authors accounted for these absorption processes in their calculations?

No, we've excluded them from the Langley-derived Io values and then interpolated with lamp-derived responsivities following the practice of Kindel et al. (2001) and of Michalsky & Kiedron (2022).

Figure 5 and Section 2.2.9: Are the authors applying the Langley-Plot method between air masses from 1.X to 6, as stated in the Figure?

Yes, the Langley's are only fit from 1.x to 6 airmasses.

Page 9, line 220: I consider it will be highlighting to include the number of Langleys performed (and maybe the time interval?). I haven't read this information in the text or maybe I have missed this number. Furthermore, the authors set a threshold in the text to define those stable Langleys performed in the whole time series: below 1% per day. Are the authors talking about the standard deviation of the fitting?

In analog to the MFRSR we apply an interquartile filter to Io values over a span of several weeks, and then apply a sliding Gaussian-weighted averaging filter to determine a daily Io. The daily Io values are stable to within a few tenths of a percent per day, which effects OD at less than 0.01.

Figure 6: Is Optical Depth (OD), TOD (as expressed in the caption) or AOD (as expressed in the legend)? Please clarify. I understand that with "good AOD", the authors are referring to those spectral bands that can be used to retrieve effectively AOD and TOD from the SAS-He. Maybe the term "good AOD" is not the best one to be included in the legend.

Clarified (lines 287-289).

Page 10, line 250: It is important to highlight that you are comparing with independent instruments!

Done (line 296).

Page 14, first paragraph: Why do you think the 1020nm spectral band presents worse results (in addition to 380nm in TRACER)?
We had noticed this issue (1020 nm) previously. However, we had not been able to identify this spectrometer degradation issue until recently. We have replaced the previous plots, which used the 1020 nm pixel from the degraded NIR spectrometer with new figures using the corresponding pixel from the UV/VIS spectrometer. Such replacement yields consistent agreement for all three campaigns. We attribute the previous disagreement at the TRACER and EPCAPE to degradation of the NIR spectrometer affecting its short wavelength range, but apparently leaving the 1.6 micron values unaffected.

In comparison with the TCAP and EPCAPE, the TRACER is characterized by slightly larger value of the RMSE (0.021) at 380 nm wavelength (Table 2). However, this value and other TRACER-related statistics (Table 2) appear feasible.